# *CHRM3 (rs2165870)* gene polymorphism is related to postoperative vomiting in female patients undergoing laparoscopic surgery. Prospective observational study

**Meng Cai◉, Lin Gan◉, Jing Li, Xiaofeng Lei, Jin Yu◉ ***

Department of Anesthesiology, Chongqing Health Center for Women and Children, Women and Children's Hospital of Chongqing Medical University, Chongqing, China

◉ These authors contributed equally to this work.
* dodoes@qq.com

**Data Availability Statement:** All relevant data are within the manuscript and its Supporting Information files.

## Abstract

### Background

Postoperative nausea and vomiting are common complications after surgery, and female patients are more likely to experience these adverse events. The goal of this study was to explore the relationship between the *CHRM3 rs2165870* polymorphism and postoperative vomiting incidence in female patients who underwent laparoscopic surgery.

### Methods

Two hundred female patients who underwent elective laparoscopic surgery with subsequent patient-controlled intravenous analgesia using dexmedetomidine and sufentanil were prospectively enrolled. The *CHRM3 rs2165870* and *KCNB2 rs349358* polymorphisms were genotyped using MassARRAY SNP typing technology. Demographic data and preoperative laboratory results of all patients were recorded. Postoperative analgesia-related information, incidence of postoperative nausea and vomiting, and other adverse events were followed up and recorded for analysis.

### Results

No significant differences were observed in any of the demographic characteristics of these patients among the different genotype carriers (*P*>0.05). The percentages of patients with each genotype of *CHRM3* were 67% (GG), 28.5% (GA) and 4.5% (AA). We found that the AA or A allele of the *CHRM3 rs2165870* polymorphism elevated the risk of postoperative vomiting (AA versus GG; OR, 6.94; 95% CI, 1.49–32.46; *P* = 0.014; A versus G; OR, 2.52; 95% CI, 1.22–5.19; *P* = 0.012). The percentages of patients with each genotype of *KCNB2* were 84.5% (TT), 15.5% (CT) and 0% (CC). There were no significant differences in the postoperative nausea or vomiting rate across the *KCNB2 rs349358* polymorphisms (*P*>0.05).

**Funding:** This study was supported by Natural Science Foundation of Chongqing of China (No. cstc2021jcyj-msxmX0763) and National Key Clinical Speciality Construction Project (Obstetrics and Gynecology).

**Competing interests:** The authors have declared that no competing interests exist.

## Conclusion

The *CHRM3 rs2165870* polymorphism is associated with the occurrence of postoperative vomiting in female patients who have undergone laparoscopic surgery. The AA genotype or A allele of the *CHRM3 rs2165870* polymorphism elevates the risk of postoperative vomiting.

## Trial registration

www.chictr.org.cn, registration number: ChiCTR2200062425.

## Introduction

Postoperative nausea and vomiting (PONV) are the most common side effects experienced after anesthesia and are also the most common complaints of surgical patients; the overall incidence of PONV is as high as 30% in the surgical population [1, 2]. Recognizing the importance of preventing and treating PONV early is essential to avoid complications after surgery and enhance patient satisfaction. Several risk factors have been identified for the occurrence of PONV, including female sex, a history of PONV/motion sickness, nonsmoking status and the use of postoperative opioids [3, 4]. In recent years, the relationship between gene polymorphisms and the occurrence of PONV has attracted increased attention. A genome-wide association study (GWAS) revealed a significant association between a single nucleotide polymorphism (SNP; *rs2165870*) located upstream of the promoter region of the M3 muscarinic acetylcholine receptor (*CHRM3*) and PONV [5, 6]. Additionally, an intergenic variant, *KCNB2 rs349358*, has also been implicated [5, 7]. The *CHRM3 rs2165870* polymorphism has been found to be associated with the efficacy of ondansetron in preventing PONV in the Chinese Han population [8]. A meta-analysis also indicated that *CHRM3 rs2165870* and the *KCNB2 rs349358* SNP appear to have significant impacts on the incidence of PONV, particularly in Caucasians [9].

Opioids are widely utilized as the primary choice for postoperative pain management [10]. In the case of female patients undergoing laparoscopy, the administration of opioids for postoperative analgesia, along with the influence of sex itself, may undeniably contribute to an increased risk of PONV. However, it remains unclear whether a correlation exists between the *CHRM3 rs2165870* and *KCNB2 rs349358* polymorphisms and the occurrence of PONV in female patients undergoing laparoscopic gynecological surgery and receiving sufentanil for postoperative pain relief. Thus, the aim of this study was to assess the impact of the *CHRM3 rs2165870* and *KCNB2 rs349358* polymorphisms on the occurrence of PONV among female patients who had undergone elective laparoscopic surgery and received patient-controlled intravenous analgesia.

## Materials and methods

This study was a prospective observation between gene polymorphisms and the occurrence of PONV among female patients. This investigation was approved by the Research Ethics Committee of the Chongqing Health Center for Women and Children (approval number: 2022–006) and was conducted in accordance with the Declaration of Helsinki. The study was registered with the Chinese Clinical Trial Registry (ChiCTR) (www.chictr.org) under registration ID ChiCTR2200062425. Written informed consent was obtained from all participants.

## Participants

The recruitment criteria included female patients aged 18 to 60 years, classified as ASA I–III, and scheduled for gynecological laparoscopic surgery under general anesthesia with the need for postoperative patient-controlled intravenous analgesia (PCIA). Furthermore, the patients were required to possess normal communication skills and the ability to appropriately use a postoperative PCIA pump. Patients with mental illness, cognitive impairments, known allergies to analgesic and sedative medications, chronic pain requiring long-term use of analgesic and sedative medications, symptoms of peripheral neuropathy, or abnormal liver and kidney function were excluded. The sample size was estimated using G Power 3.1.9.2 software. A total of 238 female patients who underwent gynecological laparoscopic surgery between August 10 2022 and December 30 2022 were enrolled in this study.

## Anesthesia and postoperative analgesia protocol

All patients underwent a standardized protocol of general intravenous anesthesia with tracheal intubation which was initiated using a controlled infusion of remifentanil (target plasma concentration of 5 ng/ml), propofol (target plasma concentration of 3.5 μg/ml), a single intravenous injection of sufentanil (0.25 μg/kg) and rocuronium (0.6 mg/kg). Anesthesia was continuously maintained using propofol and remifentanil in TCI mode, with additional doses of rocuronium administered as needed.

For PCIA, a combination of dexmedetomidine (100 μg), sufentanil (75 μg), dexamethasone (10 mg), and normal saline was used in a total volume of 100 ml. The analgesic pump used was the PCA-100C type, a disposable adjustable infusion pump manufactured by Zhejiang Chen He Medical Equipment Co., Ltd., China. The infusion was administered continuously at a rate of 2 ml/h, with a lockout time of 15 minutes. Patients were advised to self-administer a dose of 0.5 ml by pressing the control button.

## DNA extraction and genotyping

Whole venous blood samples (2 ml) were collected from each participant and stored in an EDTA anticoagulant tube. The blood was gradually cooled and then stored in a freezer at -80°C. All the extracted DNA was confirmed to meet the required criteria with a standardized DNA extraction process. MassARRAY SNP typing technology was used for SNP determination. The primer sequences used for rs2165870 were as follows: 2nd PCR reverse primer sequence 5'->3', `ACGTTGGATGAGTAGTTCTCAACAGTAGGC`; 1st PCR forward primer sequence 5'->3', `ACGTTGGATGGGAATAACAAAGCCTGGATG`; and UEP_SEQ single-base extension primer sequence, `AGTAGGCTTAAAATATTCAGTATA`. The sequences of the primers used for rs349358 were as follows: 2nd PCR reverse primer sequence 5'->3', `ACGTTGGAT GTAAGCACTTTATCCATGCCC`; 1st PCR forward primer sequence 5'->3', `ACGTTGGAT GGGCACATTACACATGTGTATG`; and UEP_SEQ single-base extension primer sequence, `AAAACACCACCTCAAAC`. After sampling, the SpectraCHIP chip was analyzed using a MALDI-TOF mass spectrometer, and the detection results were obtained using TYPER 4.0 software to obtain the raw data and genotype maps. The 384-well SpectroCHIP® bioarray chip, MassARRAY nanoinspector point prototype and MassARRAY Analyzer 4.0 mass spectrometer were obtained from Agena Bioscience, Inc.

## Observation indicators

Demographics and preoperative laboratory examinations of all participants were recorded. The incidences of postoperative nausea and vomiting, as well as other adverse events

(hypotension, respiratory depression, bradycardia, dizziness, drowsiness, urinary retention and chills), were followed up and recorded by an anesthesiologist for analysis. Postoperative analgesic-related information, including the number of self-controlled compressions performed with the analgesic pump within the first 24 hours postoperatively, the time when the analgesics in the pump were completely depleted, the dosage of the postoperative 24-hour analgesic solution, the longest continuous sleep time on the night of the operation, the time to first mobilization out of bed and the time to first flatus after the operation, the duration of urinary catheterization and the number of postoperative hospitalization days, were also recorded.

### Definition of adverse events

Hypotension was defined as a systolic blood pressure less than 90 mmHg; respiratory depression was defined as $SpO_2$ <90% under inhalation of air; bradycardia was defined as a heart rate of <50 beats/minute; nausea was defined as discomfort in the upper abdomen and a feeling of urgency to vomit; vomiting was defined as the phenomenon of ejecting the contents of the stomach to pass through the esophagus and mouth due to the strong contraction of the stomach; and urinary retention was defined as the inability to discharge urine when the bladder was filled with urine, resulting in unbearable bloating and pain. Dizziness, drowsiness and chills were recorded based on the patient's subjective statement.

### Sample size calculation

Based on data from the literature [11, 12], the baseline incidence rate of PONV after such surgeries is estimated to be 30–50%. Based on the mutation proportion of CHRM3 [11], it is expected that the proportions of the GG, GA, and AA genotypes are 45%, 45%, and 10%, respectively. G Power 3.1.9.2 software was used to determine the sample size before this study under the following conditions: $\alpha = 0.05$, $\beta = 0.2$, power value = 0.8, and estimated odds ratio (OR) = 2.79. The total sample size required was 167.

### Statistical methods

All the statistical analyses were performed with SPSS 21.0 (SPSS, Inc., Chicago, USA). Hardy–Weinberg equilibrium among groups was assessed using the $\chi 2$ test. One-way ANOVA for homogeneous variances was applied for comparison of measurement data between groups, and Brown-Forsythe ANOVA was used for heterogeneous variances. The chi-square test and Fisher's exact test were used to determine differences in the incidence of postoperative adverse reactions. A nonparametric test (Kruskal–Wallis rank test) was used for normally distributed data. The crude odds ratios (ORs) and 95% confidence intervals (CIs) were evaluated via logistic regression analysis. A *P* value < 0.05 was considered to indicate statistical significance.

## Results

### Study population

Two hundred and thirty-eight female patients were enrolled in the study, and two hundred patients were ultimately analyzed (**Fig 1**). No significant differences were observed in any of the demographic data and preoperative laboratory examination results of these patients among the different genotype carriers (*P*>0.05) (**Tables 1 and 2**). The percentages of patients with each genotype of *CHRM3* were 67% (GG), 28.5% (GA), and 4.5% (AA). The distribution of the three genotypes conformed to Hardy–Weinberg equilibrium after the performance of a chi-square test ($\chi 2 = 0.401$, *P* value = 0.818>0.05). The percentages of patients with each genotype of *KCNB2* were 84.5% (TT), 15.5% (CT), and 0% (CC) (**Fig 2**).

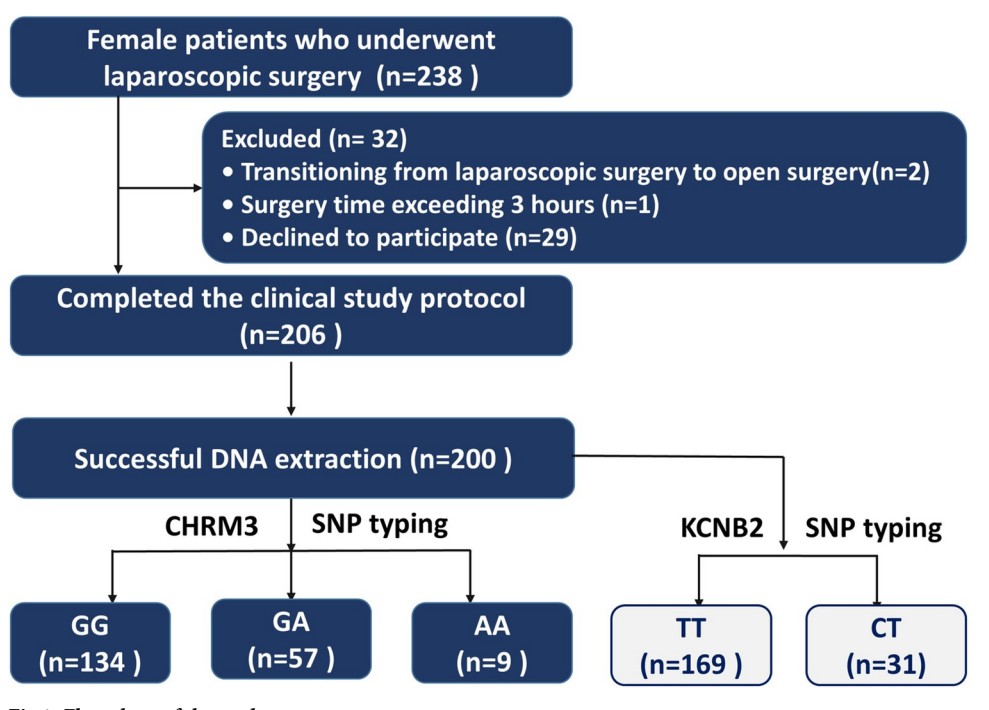

**Fig 1. Flow chart of the study.**

## Postoperative analgesia-related information

There were no significant differences in postoperative analgesia-related indicators across the three SNP loci of the *CHRM3* (*P*>0.05), including the number of self-controlled compressions with the analgesic pump within the first 24 hours postoperatively, the time when the analgesics in the pump were completely depleted, the dosage of the postoperative 24-hour analgesic solution, the longest continuous sleep time on the night of the operation, the time to start mobilizing out of bed and the time to first flatus after the operation, the duration of urinary catheterization, or the postoperative hospitalization days (**Table 3**).

**Table 1. Demographics and preoperative laboratory examination results for the *CHRM3* rs2165870 genotype.**

| | GG (n = 134) | GA (n = 57) | AA (n = 9) | *P* Value |
|---|---|---|---|---|
| Total number (%) | 134 (67%) | 57 (28.5%) | 9 (4.5%) | / |
| Age, years | 39.99±10.17 | 37.79±8.51 | 38.67±10.74 | 0.358 |
| Weight, kg | 57.53±9.18 | 57.25±6.31 | 57.22±5.07 | 0.975 |
| Height, cm | 159.11±5.01 | 158.27±4.34 | 159.56±5.32 | 0.505 |
| BMI, kg/m$^2$ | 22.78±3.44 | 23.06±3.16 | 22.54±2.44 | 0.829 |
| Hemoglobin, g/L | 126.75±16.93 | 123.47±13.33 | 129.67±8.31 | 0.323 |
| Hematocrit, % | 39.07±4.05 | 38.25±3.60 | 39.31±2.16 | 0.381 |
| Leukocytes, x 10$^9$/L | 5.47±1.31 | 5.94±1.62 | 5.17±1.55 | 0.128 |
| Platelets, x 10$^9$/L | 231.39±61.36 | 234.95±69.92 | 243.00±59.76 | 0.836 |
| Duration of the operation, min | 106.96±44.26 | 109.86±53.85 | 88.67±28.26 | 0.450 |
| Previous surgeries (Yes/No) | 76/58 | 32/25 | 8/1 | 0.158 |
| Previous history of pregnancy (Yes/No) | 93/41 | 40/17 | 7/2 | 0.868 |
| Previous history of delivery history (Yes/No) | 81/53 | 35/22 | 5/4 | 0.946 |

The data are presented as means ±SDs

**Table 2. Demographics and preoperative laboratory examination results for *KCNB2 rs349358* genotype.**

| | TT (n = 169) | CT (n = 31) | *P* Value |
|---|---|---|---|
| Total number (%) | 169 (84.5%) | 31 (15.5%) | / |
| Age, year | 38.99±9.75 | 40.97±9.76 | 0.302 |
| Weight, kg | 57.41±8.29 | 57.56±8.37 | 0.853 |
| Height, cm | 158.92±4.84 | 158.74±4.84 | 0.926 |
| BMI, kg/m$^2$ | 22.83±3.34 | 22.95±3.23 | 0.859 |
| Hemoglobin, g/L | 126.01±16.08 | 125.65±13.82 | 0.907 |
| Hematocrit, % | 38.84±3.95 | 38.84±3.41 | 0.995 |
| Leukocyte, x 10$^9$/L | 5.64±1.42 | 5.30±1.44 | 0.226 |
| Platelet, x 10$^9$/L | 234.50±65.51 | 224.32±51.89 | 0.414 |
| Duration of operation, min | 105.97±44.97 | 112.35±55.37 | 0.485 |
| Previous surgeries (Yes/No) | 98 / 71 | 18 / 13 | 0.994 |
| Previous pregnant history (Yes/No) | 117 / 52 | 23 / 8 | 0.579 |
| Previous deliver history (Yes/No) | 101 / 68 | 20 / 11 | 0.619 |

The data are presented as means ±SDs.

## Association of the *CHRM3 rs2165870* polymorphism with postoperative adverse reactions

The incidences of nausea during postoperative analgesia were 12.7% with GG, 17.5% with GA and 33.3% with AA, and the incidences of vomiting during postoperative analgesia were 6.7%

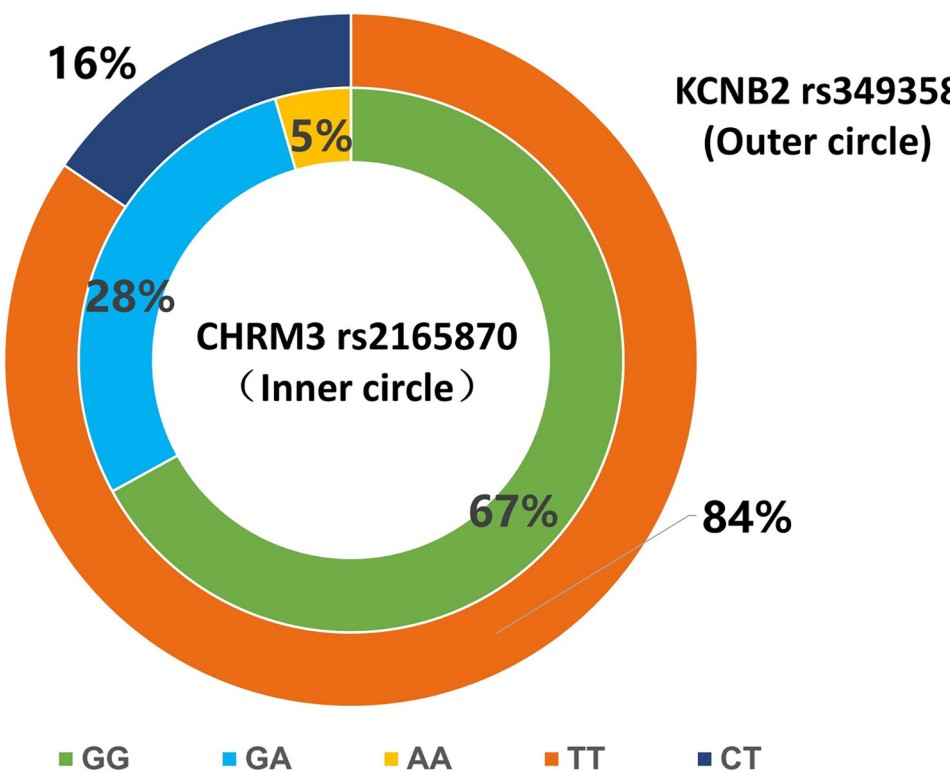

**Fig 2. Genotype distribution composition ratio.** The inner circle represents *CHRM3* rs2165870, and the outer circle represents *KCNB2 rs349358*.

**Table 3. Postoperative analgesic-related information for the *CHRM3* rs2165870 genotype.**

| Parameters | GG (n = 134) | GA (n = 57) | AA (n = 9) | *P* Value |
|---|---|---|---|---|
| Number of self-controlled compressions of the with analgesic pump within 24 hours postoperatively | 2.38±4.52 0 (0, 3) | 1.11±1.89 0 (0, 1.5) | 2.00±2.78 0 (0, 5.5) | 0.119 |
| The time when the analgesics in the pump were completely depleted (hours) | 49.51±1.82 50 (49, 50) | 49.67±0.50 50 (49.31, 50) | 49.53±0.62 50 (48.88, 50) | 0.806 |
| Postoperative 24-hour analgesic solution dosage (ml) | 49.51±1.82 48 (48, 49.5) | 48.22±3.55 48 (48, 49) | 49.00±1.39 48 (48, 50.75) | 0.913 |
| The longest continuous sleep time on the night of the operation (min) | 285.67±102.84 240 (90, 300) | 286.32±108.03 300 (225, 360) | 213.33±112.69 300 (240, 360) | 0.130 |
| Time to start moving out of bed after the operation(hours) | 21.49±4.54 21 (18, 24) | 21.46±5.47 22 (18, 24.15) | 19.36±3.49 18 (17, 21.75) | 0.429 |
| Time to first flatus after operation (hours) | 21.85±6.37 21.5 (17.5, 26) | 21.80±5.28 22 (17, 25.65) | 20.08±9.17 20 (14, 25.75) | 0.710 |
| Duration of urinary catheterization (hours) | 24.35±10.67 21 (18, 25) | 24.69±9.97 22 (18, 26) | 23.79±11.12 18.6 (16.5, 33.25) | 0.962 |
| Length of postoperative hospitalization (days) | 4.68±2.60 4 (4, 5) | 4.68±2.29 4 (3, 5) | 5.00±1.12 5 (4, 6) | 0.931 |

The data are presented as means ±SDs or as medians (quartile)

with GG, 12.3% with GA and 33.3% with AA. The incidence of vomiting during postoperative analgesia in patients with homozygous AA mutations was greater than that in patients with GG mutations ($P<0.05$). There were no significant differences in the incidence of other adverse events across the three genotypes of the *CHRM3* SNP ($P>0.05$) (**Table 4**).

The AA genotype was related to an elevated risk of postoperative vomiting (AA versus GG; OR, 6.94; 95% CI, 1.49–32.46; $P = 0.014$). In addition, the A allele of the *CHRM3 rs2165870* polymorphism increased the risk of postoperative vomiting according to the allelic model (A versus G; OR, 2.52; 95% CI, 1.22–5.19; $P = 0.012$) (**Table 5**).

## Association of the *KCNB2 rs349358* polymorphism with postoperative analgesia-related information and adverse reactions

There were no significant differences observed in postoperative analgesia-related indicators mentioned above or adverse reaction rates across the *KCNB2 rs349358* polymorphism ($P>0.05$). The incidence of nausea during postoperative analgesia was 19.35% with CT and

**Table 4. Comparison of postoperative adverse reactions among patients with different CHRM3 genotypes.**

| Adverse reactions | GG (n = 134) | GA (n = 57) | AA (n = 9) | *P* Value |
|---|---|---|---|---|
| Hypotension, n (Yes/No) | 4/130 | 0/57 | 0/9 | 0.366 |
| Respiratory depression, n (Yes/No) | 0/134 | 0/57 | 0/9 | / |
| Bradycardia, n (Yes/No) | 5/129 | 0/57 | 0/9 | 0.283 |
| Nausea, n (Yes/No) | 17/117 | 10/47 | 3/6 | 0.199 |
| Vomit, n (Yes/No) | 9/125 | 7/50 | 3/6* | 0.022 |
| Dizzy, n (Yes/No) | 4/130 | 1/56 | 0/9 | 0.783 |
| Drowsiness, n (Yes/No) | 0/134 | 0/57 | 0/9 | / |
| Urinary retention, n (Yes/No) | 1/133 | 1/56 | 0/9 | 0.777 |
| Chills, n (Yes/No) | 7/127 | 4/53 | 0/9 | 0.672 |

* compared with the GG group, $P < 0.05$.

**Table 5. Postoperative vomiting occurrence in patients with the *CHRM3* genotypes.**

| Models | Genotype | Vomiting Cases (n = 19) (n, %) | No-Vomiting Cases (n = 181) (n, %) | OR (95% CI) | *P* value |
|---|---|---|---|---|---|
| Codominant | GG | 9 (47.37%) | 125 (69.06%) | 1.00 (reference) | / |
| Heterozygote | GA | 7 (36.84%) | 50 (27.62%) | 1.94 (0.69,5.51) | 0.21 |
| Homozygote | AA | 3 (15.79%) | 6 (3.31%) | 6.94 (1.49,32.46) | 0.014 |
| Dominant | GG | 9 (47.37%) | 125 (69.06%) | 1.00 (reference) | / |
| | GA+AA | 10 (52.63%) | 56 (30.94%) | 2.48 (0.96, 6.44) | 0.062 |
| Recessive | GG+GA | 16 (84.21%) | 175 (96.69%) | 1.00 (reference) | / |
| | AA | 3 (15.79%) | 6 (3.31%) | 5.47 (1.25,23.96) | 0.024 |
| Allele | G | 25 (65.79%) | 300 (82.87%) | 1.00 (reference) | / |
| | A | 13 (34.21%) | 62 (17.13%) | 2.52 (1.22, 5.19) | 0.012 |

14.20% with TT, and the incidence of vomiting was 9.68% with CT and 9.47% with TT (*P*>0.05) (**Tables 6 and 7**).

## Discussion

PONV is a common complication following surgical procedures that can significantly impact patient outcomes and satisfaction [1, 2]. Genetic variations such as *CHRM3 rs2165870* have been suggested to play a role in the pathogenesis of PONV [8]. This study confirmed that the *CHRM3 rs2165870* polymorphism is related to the incidence of postoperative vomiting in female patients undergoing laparoscopic surgery. The AA genotype or A allele of the *CHRM3 rs2165870* polymorphism was found to elevate the risk of postoperative vomiting.

The *CHRM3* gene encodes the muscarinic acetylcholine receptor subtype 3 (M3 receptor), which is primarily expressed in the central nervous system and peripheral tissues, including the gastrointestinal tract [13]. The M3 receptor is a G protein-coupled receptor involved in the transmission of cholinergic signals and plays a crucial role in regulating smooth muscle contraction, salivation, and gastric acid secretion. The *rs2165870* SNP involves the substitution of

**Table 6. Postoperative analgesic-related information for the *KCNB2 rs349358* genotype.**

| Parameters | TT (n = 169) | CT (n = 31) | *P* Value |
|---|---|---|---|
| Numbers of self-control compressions with analgesic pump within postoperative 24 hours. | 2.04±4.11<br>0 (0, 2) | 1.77±2.66<br>0 (0, 2.5) | 0.728 |
| The time when analgesics in pump is completely depleted (hour) | 49.56±1.60<br>50 (49, 50) | 49.53±0.98<br>50 (49, 50) | 0.941 |
| Postoperative 24-hour analgesic solution dosage (ml) | 48.29±5.65<br>48.5 (48, 49.5) | 47.79±6.39<br>48 (48, 49.5) | 0.660 |
| The longest continuous sleep time on the night of operation (min) | 282.31±103.47<br>300 (180, 360) | 284.19±116.64<br>300 (240, 360) | 0.927 |
| Time to start moving out of bed after operation (hour) | 21.43±4.94<br>21 (18, 24) | 21.14±3.93<br>21 (18, 24) | 0.756 |
| Time to start farting after operation (hour) | 21.69±6.11<br>22 (17.5, 27) | 22.13±6.78<br>21.5 (17, 26) | 0.718 |
| Duration of urinary catheter (hour) | 24.28±10.50<br>21 (18.7, 27) | 25.19±10.27<br>21 (18, 24.6) | 0.656 |
| Postoperative hospitalization days (day) | 4.78±2.57<br>4 (3, 5) | 4.26±1.73<br>4 (4, 5) | 0.283 |

Data is presented as Mean ±SD or Median (quartile)

**Table 7. Comparison of postoperative adverse reactions among patients with different *KCNB2 rs349358* genotypes.**

| | TT (n = 169) | CT (n = 31) | *P* Value |
|---|---|---|---|
| Hypotension, n (Yes/No) | 2 / 167 | 2 / 29 | 0.054 |
| Respiratory depression, n (Yes/No) | 0 /169 | 0 / 31 | / |
| Bradycardia, n (Yes/No) | 4 / 165 | 1 / 30 | 0.778 |
| Nausea, n (Yes/No) | 24 / 145 | 6 / 25 | 0.460 |
| Vomit, n (Yes/No) | 16 / 153 | 3 / 28 | 0.971 |
| Dizzy, n (Yes/No) | 5 / 164 | 0 / 31 | 0.427 |
| Drowsiness, n (Yes/No) | 0 / 169 | 0 / 31 | / |
| Urinary retention, n (Yes/No) | 1 / 168 | 1 / 30 | 0.287 |
| Chills, n (Yes/No) | 9 / 160 | 2 / 29 | 0.800 |

adenine (A) for guanine (G) at position 1358 of the *CHRM3* gene. The different genotypes associated with this polymorphism are GG, AG, and AA, representing the homozygous wild-type, heterozygous and homozygous variant genotypes, respectively [13]. Previous studies demonstrated that sequence variations in the promoter region of the *CHRM3* gene may be associated with asthma, atopy and early-onset type 2 diabetes [14, 15].

A GWAS of 122 surgery patients with severe PONV and 129 matched controls revealed that the GG+AA genotype of the *CHRM3 rs2165870* polymorphism was the only risk factor related to PONV. Moreover, the A allele of the *CHRM3 rs2165870* polymorphism elevated the risk of PONV (OR = 2.3, 95% CI = 1.36–4) [5]. Another prospective, controlled study with 454 individuals undergoing elective surgery reported that the *CHRM3 rs2165870* polymorphism could predict PONV susceptibility and that AA or GA genotype carriers had a higher risk for PONV than GG genotype carriers [6]. A retrospective study involving 472 patients who underwent elective surgery also revealed that the *CHRM3 rs2165870* and *KCNB2 rs349358* polymorphisms increased the risk of PONV [7]. In this study, we observed that the AA or A allele of the *CHRM3 rs2165870* polymorphism increased the risk of postoperative vomiting in female patients who underwent laparoscopic surgery with PCIA, which is consistent with the findings of the above mentioned studies. However, in this study, there was no significant difference in the risk of postoperative nausea, although it differed during postoperative analgesia. The possible reason may be the clinical heterogeneity of these disorders, the different sample sizes or the differences in the inclusion criteria of patients. Notably, no association between *KCNB2 rs349358* and PONV was found in this study.

The variant A allele of the *CHRM3* polymorphism may confer increased susceptibility to PONV, possibly through dysregulated cholinergic signaling in the central and peripheral nervous systems. Cholinergic signaling plays a crucial role in the regulation of nausea and vomiting pathways. Stimulation of M3 receptors in the postrema area, which is a key emetic center in the brain, may trigger the emetic reflex. Altered M3 receptor function due to the variant A allele may disrupt the normal balance of cholinergic signaling, resulting in increased susceptibility to PONV. Furthermore, the M3 receptor is also present in the gastrointestinal tract, where it regulates smooth muscle contraction and gut motility. Dysfunctional M3 receptors due to the variant A allele may impair gastric emptying and intestinal transit [16, 17], contributing to the development of PONV.

Patients with the AA genotype or who were A allele carriers had a high incidence of PONV after elective surgery, which encouraged clinicians to use medications for these patients to prevent the occurrence of PONV. Thus, genetic analysis of surgical patients before anesthesia may be necessary in the future.

This study has several limitations. Despite the promising findings regarding the association between the *CHRM3 rs2165870* genotype and postoperative vomiting in female patients undergoing laparoscopic surgery, it is important to note that the sample size in our study was relatively small, especially because the AA genotype had fewer mutations than estimated. Therefore, well-designed studies to confirm these associations are needed. Additionally, the genetic basis of PONV is likely multifactorial and involves the interplay of multiple genes and environmental factors. It is thus necessary to conduct research with larger sample sizes and explore the feasibility of providing individual preventive treatment for patients with the A allele to provide a more comprehensive understanding of this phenomenon.

## Conclusions

This study revealed that the *CHRM3 rs2165870* polymorphism, rather than *KCNB2*, is related to postoperative vomiting in female patients undergoing laparoscopic surgery. The AA genotype or A allele of the *CHRM3 rs2165870* polymorphism elevates the risk of postoperative vomiting.

## Supporting information

**S1 Checklist. STROBE statement—Checklist of items that should be included in reports of *cohort studies*.**
(DOCX)

**S2 Checklist. TREND statement checklist.**
(PDF)

**S1 Protocol.**
(DOCX)

## Acknowledgments

We are grateful to Prof. Yongchun Su at Chongqing Youyoubaobei Women and Children's Hospital for his manuscript preparation assistance.

## Author Contributions

**Conceptualization:** Meng Cai, Jin Yu.

**Data curation:** Jin Yu.

**Formal analysis:** Lin Gan, Jing Li.

**Funding acquisition:** Jin Yu.

**Investigation:** Lin Gan, Jing Li.

**Methodology:** Meng Cai.

**Resources:** Xiaofeng Lei.

**Supervision:** Xiaofeng Lei.

**Validation:** Jing Li.

**Writing – original draft:** Lin Gan.

**Writing – review & editing:** Jin Yu.

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
