## [Decision Letter · Decision Letter 0]

8 May 2024

PONE-D-24-07292CHRM3 (rs2165870) gene polymorphism is related to postoperative vomiting in female patients undergoing laparoscopic surgeryPLOS ONE

Dear Dr. Yu,

Thank you for submitting your manuscript to PLOS ONE. After careful consideration, we feel that it has merit but does not fully meet PLOS ONE’s publication criteria as it currently stands. Therefore, we invite you to submit a revised version of the manuscript that addresses the points raised during the review process.Although I found this a very interesting study and a well written manuscript, I would suggest that you answer all the questions raised by the reviewers, in particular reviewer #3.

I look forward to receive clear and "track-change" copies of your reviewed manuscript.

We look forward to receiving your revised manuscript.

Kind regards,

Guilherme Antonio Moreira de Barros, M.D., M.Sc., Ph.D

Academic Editor

PLOS ONE

Journal Requirements:

A clean copy of the edited manuscript (uploaded as the new *manuscript* file)”.

 [This study was supported by Natural Science Foundation of Chongqing of China (No. cstc2021jcyj-msxmX0763) and National Key Clinical Speciality Construction Project (Obstetrics and Gynecology).].  

Reviewers' comments:

Reviewer's Responses to Questions

**Comments to the Author**

1. Is the manuscript technically sound, and do the data support the conclusions?

Reviewer #1: Yes

Reviewer #2: Yes

Reviewer #3: No

2. Has the statistical analysis been performed appropriately and rigorously? 

Reviewer #1: Yes

Reviewer #2: N/A

Reviewer #3: No

3. Have the authors made all data underlying the findings in their manuscript fully available?

Reviewer #1: Yes

Reviewer #2: No

Reviewer #3: Yes

4. Is the manuscript presented in an intelligible fashion and written in standard English?

Reviewer #1: Yes

Reviewer #2: Yes

Reviewer #3: Yes

5. Review Comments to the Author

Reviewer #1: To Authors:

The authors aimed to explore the relationship between the CHRM3 rs2165870 polymorphism and postoperative vomiting incidence in female patients who underwent laparoscopic surgery, resulting in the outcomes that the AA or A allele of the CHRM3 rs2165870 polymorphism elevated the risk of postoperative vomiting (AA versus GG; OR, 6.94; 95% CI, 1.49-32.46; P = 0.014; A versus G; OR, 2.52; 95% CI, 1.22-5.19; P = 0.012). The manuscript is largely well written and informative overall. However, there seem to be only a few minor concerns in this manuscript. The paper will be improved when the authors revise them according to the following comments:

[Minor points]

All manuscript and supporting information:

The gene names should be italic throughout the manuscript and supporting information.

Results:

“the three SNP loci of CHRM3” should be changed to “the three genotypes of the CHRM3 SNP”. [Page 11, Lines 204-205]

Results:

It would be better to describe the details about the results of association analysis between the postoperative analgesia-related indicators or adverse reaction rates between the KCNB2 rs349358 genotypes as well as CHRM3 rs2165870 genotypes. Even if such details are not presented in a main table, it would be better to be presented in a table in supporting information.

Table 3:

“P” in the “P < 0.05” should be italic.

Reviewer #2: Title: Authors should include the type of study along with the title.

CHRM3 (rs2165870) gene polymorphism is related to postoperative vomiting in female patients undergoing laparoscopic surgery. Prospective observational study.

Abstract- Before the objectives, I missed a short background.

Introduction – ok

Methods:

Explain how and why the sample size applied was stated. Explain how the study size was arrived at.

Address possible confounding factors. And how it would alter the results.

Any missing data? And how do you address it?

Discuss the generalisability of the results.

Reviewer #3: The literature review needs to thoroughly cover other adverse events. The background does not provide enough justification as to why vomiting is considered more important than other adverse events.

The sample size calculation lacks clarity. Stating the software used is not sufficient. It is necessary to clarify the assumptions and tests employed. Regarding the current results, it is difficult to determine whether the nonsignificant findings are due to insufficient statistical power.

The statistical analysis used to evaluate the association between genotypes and adverse events appears to be cherry-picking, as the only significance is for vomiting. Nausea was not discussed in the background. P-values need to be adjusted for the 9 tests conducted (Table 3), and a marginally significant p-value of 0.022 will no longer be significant after adjustment.

Further, is the marginal significance driven by the rare allele (AA, n=9)? The incident rates of vomiting for GG, GA and AA are 7%, 12% and 30% respectively and the significance is between GG and AA but not between GA and AA. Is the significance attributed to random chance? If the sample size is not determined correctly, the results will be inconclusive.

6. PLOS authors have the option to publish the peer review history of their article (what does this mean?). If published, this will include your full peer review and any attached files.

Reviewer #1: No

Reviewer #2: No

Reviewer #3: No

---

## [Author Response · Author response to Decision Letter 0]

6 Jun 2024

June 7, 2024 

Guilherme Antonio Moreira de Barros

Academic Editor

PLOS ONE

Re: [PONE-D-24-07292

CHRM3 (rs2165870) gene polymorphism is related to postoperative vomiting in female patients undergoing laparoscopic surgery]

Dear Editor Guilherme Antonio Moreira de Barros, 

Thank you for your letter dated May 9, 2024, regarding our manuscript. We appreciate the helpful comments from the editor and reviewers. We have responded to these comments in detail below (the original criticisms from the reviewers are in italics). We revised the manuscript with tracked changes indicating all modifications which can be seen in the resubmitted version (please refer to the Word document).

Response to Reviewer 1

1.The authors aimed to explore the relationship between the CHRM3 rs2165870 polymorphism and postoperative vomiting incidence in female patients who underwent laparoscopic surgery, resulting in the outcomes that the AA or A allele of the CHRM3 rs2165870 polymorphism elevated the risk of postoperative vomiting (AA versus GG; OR, 6.94; 95% CI, 1.49-32.46; P = 0.014; A versus G; OR, 2.52; 95% CI, 1.22-5.19; P = 0.012). The manuscript is largely well written and informative overall. However, there seem to be only a few minor concerns in this manuscript. The paper will be improved when the authors revise them according to the following comments

Response: Thank you very much for taking the time to review our manuscript. We sincerely appreciate the effort and thought you put into providing such valuable feedback. The positive comments means a lot to us. 

2.All manuscript and supporting information: The gene names should be italic throughout the manuscript and supporting information.

Response: Thank you for this comment. We have changed all gene names to italic format. Please see the revised manuscript for details.

3. Results:

“the three SNP loci of CHRM3” should be changed to “the three genotypes of the CHRM3 SNP”. [Page 11, Lines 204-205]

Response: Thank you for the reminder. We have made this change in the revised manuscript. 

4. Results:

It would be better to describe the details about the results of association analysis between the postoperative analgesia-related indicators or adverse reaction rates between the KCNB2 rs349358 genotypes as well as CHRM3 rs2165870 genotypes. Even if such details are not presented in a main table, it would be better to be presented in a table in supporting information.

Response: Thank you for your comment. We have added relevant information about KCNB2 rs349358 genotypes and provided them as Supplementary Table1-3. 

5. Table 3:

“P” in the “P < 0.05” should be italic.

Response: Thank you for the reminder. We have made this change in the revised manuscript. 

Response to Reviewer 2

1. Title: Authors should include the type of study along with the title.

CHRM3 (rs2165870) gene polymorphism is related to postoperative vomiting in female patients undergoing laparoscopic surgery. Prospective observational study.

Response: Thank you for your review and comments. This is a very helpful suggestion, and we have changed the manuscript's title as your suggestion.

2. Abstract- Before the objectives, I missed a short background.

Introduction – ok

Response: We appreciate your feedback. In the abstract, we have added a brief background before the objectives. 

3. Methods:

Explain how and why the sample size applied was stated. Explain how the study size was arrived at.

Address possible confounding factors. And how it would alter the results.

Any missing data? And how do you address it?

Response: We appreciate your remarks. The updated manuscript includes further information about the basis for the sample size estimate. Based on data from the literature (PMID: 18633022, PMID: 29843509), the baseline incidence rate of PONV after such surgeries is estimated to be 30-50%. Based on the mutation proportion of CHRM3 (PMID: 18633022), it is expected that the proportions of GG, GA, and AA genotypes are 45%, 45%, and 10%, respectively. Software G Power 3.1.9.2 was used to determine the sample size before this study with the following conditions: α = 0.05, β = 0.2, power value = 0.8, and the estimated odds ratio (OR) = 2.79. The total sample size required is 167.

Possible confounding factors in this study including population stratification, linkage disequilibrium, environmental and lifestyle factors, measurement error and sample size etc. may all affect the results. Especially, the actual mutation rates were 67% (GG), 28.5% (GA), and 4.5% (AA) which is extremely different as initial estimated. Therefore, we recognized that increased sample size would enhance the statistical power.

Furthermore, we missed detailed data related to KCNB2 in the initial submitted manuscript. It was not shown in the first version of the manuscript due to the lack of significant differences. And it has been included in the supplemental materials in the revised manuscript version.

4. Discuss the generalisability of the results.

Response: Thank you for this comment. We have added the generalizability of the results in the discussion. The results indicates that patients with specific gene polymorphism should be paid more attention and may be given certain medications to prevent the occurrence of PONV. 

Response to Reviewer 3

1. The literature review needs to thoroughly cover other adverse events. The background does not provide enough justification as to why vomiting is considered more important than other adverse events.

Response: We appreciate your careful and considerate review remarks. In fact, all related adverse events beside nausea and vomiting were investigated and analyzed in the study and there is no significant difference among different genotypes of focused two genes. Vomiting was paid more attention in this study because it is the most often reported postoperative complication following laparoscopic surgery by patients. 

2. The sample size calculation lacks clarity. Stating the software used is not sufficient. It is necessary to clarify the assumptions and tests employed. Regarding the current results, it is difficult to determine whether the nonsignificant findings are due to insufficient statistical power.

Response: Thank you for the comment. We acknowledge that we should have included more information about the sample size calculation in the text. In the revised manuscript, we have added the relevant information on sample size calculation as blows. 

Based on data from the literature (PMID: 18633022, PMID: 29843509), the baseline incidence rate of PONV after such surgeries is estimated to be 30-50%. Based on the mutation proportion of CHRM3 (PMID: 18633022), it is expected that the proportions of GG, GA, and AA genotypes are 45%, 45%, and 10%, respectively. Software G Power 3.1.9.2 was used to determine the sample size before this study with the following conditions: α = 0.05, β = 0.2, power value = 0.8, and the estimated odds ratio (OR) = 2.79. The total sample size required is 167.

3. The statistical analysis used to evaluate the association between genotypes and adverse events appears to be cherry-picking, as the only significance is for vomiting. Nausea was not discussed in the background. P-values need to be adjusted for the 9 tests conducted (Table 3), and a marginally significant p-value of 0.022 will no longer be significant after adjustment.

Response: 

Thank you for the comments. Based on the data obtained in this study, the significant difference of adverse reactions only found in vomiiting and not in nausea among different CHRM3 genotypes. And there is not significant difference both in nausea and vomiting among KCNB2 rs349358 genotypes by statistical analysis. For CHRM3 genotypes, the final column of Table 3's P-value indicates the total P-value for the comparison of the three groups. The P-values for pairwise comparisons between the three groups for the individual indicator of vomiting are as follows.

Group P value

GG Vs.GA 0.204

GG Vs. AA 0.005

AA Vs.GA 0.102

Actually and same as your comments, the statistical difference was found between GG and AA genotype of gene CHRM3. The overall comparison p-value of vomiting incidence among the three groups is 0.022.

Moreover, the allelic model (A versus G; OR, 2.52; 95% CI, 1.22-5.19; P = 0.012) revealed that the A allele of the CHRM3 rs2165870 polymorphism raised the risk of postoperative vomiting in additional analysis. Therefore, despite the limited sample size, the current results show that the CHRM3 rs2165870 polymorphism's A allele increased the risk of postoperative vomiting, and this finding merits further investigation in clinical practice.

4. Further, is the marginal significance driven by the rare allele (AA, n=9)? The incident rates of vomiting for GG, GA and AA are 7%, 12% and 30% respectively and the significance is between GG and AA but not between GA and AA. Is the significance attributed to random chance? If the sample size is not determined correctly, the results will be inconclusive.

Response: We appreciate the insightful comments on the study's sample size. We are grateful for your concern and would like to go over this in more detail. As can be seen in the revised manuscript, the sample size was decided upon a thorough statistical power analysis which is sufficient to detect the expected effect size with a power of 0.8 at a significance level of 0.05. 

Initially, we roughly estimated that the GG, GA, and AA genotypes mutation rates is 45%, 45%, and 10%, respectively. The actual mutation rates, however, were found to be 4.5% (AA), 28.5% (GA), and 67% (GG). Exactly there are fewer mutations in the AA genotype than estimated. The incidence rates of vomiting for GG, GA, and AA are indeed 7%, 12%, and 30%, respectively. The observed significance between GG and AA, but not between GA and AA, suggests that the difference was not be solely due to random chance.

However, we acknowledged the possibility that a larger sample size maybe yield a greater statistical power or even different statistical results.

Thanks again, we appreciate your suggestion, and we have addressed the sample size limitations in the revised manuscript.

Sincerely,

Dr. Yu Jin on behalf of all authors.

Department of Anesthesiology, 

Chongqing Health Center for Women and Children, Women and Children’s Hospital of Chongqing Medical University, Chongqing, China, 401147#

---

## [Editor Report · Decision Letter 1]

18 Jun 2024

PONE-D-24-07292R1CHRM3 (rs2165870) gene polymorphism is related to postoperative vomiting in female patients undergoing laparoscopic surgery. Prospective observational studyPLOS ONE

Dear Dr. Yu,

Thank you for submitting your manuscript to PLOS ONE. After careful consideration, we feel that it has merit but does not fully meet PLOS ONE’s publication criteria as it currently stands. Therefore, we invite you to submit a revised version of the manuscript that addresses the points raised during the review process.

We look forward to receiving your revised manuscript.

Kind regards,

Guilherme Antonio Moreira de Barros, M.D., M.Sc., Ph.D

Academic Editor

PLOS ONE

Additional Editor Comments:

The authors aimed to explore the relationship between the CHRM3 rs2165870 polymorphism and postoperative vomiting incidence in female patients who underwent laparoscopic surgery, resulting in the outcomes that the AA or A allele of the CHRM3 rs2165870 polymorphism elevated the risk of postoperative vomiting (AA versus GG; OR, 6.94; 95% CI, 1.49-32.46; P = 0.014; A versus G; OR, 2.52; 95% CI, 1.22-5.19; P = 0.012). The manuscript is largely well written and informative overall. However, there seem to be only a few minor concerns in this manuscript. The paper will be improved when the authors revise them according to the following comments:

[Minor points]

1. All manuscript and supporting information:

The gene names should be italic throughout the manuscript and supporting information.

2. Results:

“the three SNP loci of CHRM3” should be changed to “the three genotypes of the CHRM3 SNP”. [Page 11, Lines 204-205]

3. Results:

It would be better to describe the details about the results of association analysis between the postoperative analgesia-related indicators or adverse reaction rates between the KCNB2 rs349358 genotypes as well as CHRM3 rs2165870 genotypes. Even if such details are not presented in a main table, it would be better to be presented in a table in supporting information.

4. Table 3:

“P” in the “P < 0.05” should be italic.

------------

5. Abstract- Before the objectives, I missed a short background.

6. Explain how and why the sample size applied was stated. Explain how the study size was arrived at.

7. Address possible confounding factors. And how it would alter the results.

8. Any missing data? And how do you address it?

9. Discuss the generalisability of the results.

------------

10. The literature review needs to thoroughly cover other adverse events. The background does not provide enough justification as to why vomiting is considered more important than other adverse events.

11. The sample size calculation lacks clarity. Stating the software used is not sufficient. It is necessary to clarify the assumptions and tests employed. Regarding the current results, it is difficult to determine whether the nonsignificant findings are due to insufficient statistical power.

12. The statistical analysis used to evaluate the association between genotypes and adverse events appears to be cherry-picking, as the only significance is for vomiting. Nausea was not discussed in the background. P-values need to be adjusted for the 9 tests conducted (Table 3), and a marginally significant p-value of 0.022 will no longer be significant after adjustment.

13. Further, is the marginal significance driven by the rare allele (AA, n=9)? The incident rates of vomiting for GG, GA and AA are 7%, 12% and 30% respectively and the significance is between GG and AA but not between GA and AA. Is the significance attributed to random chance? If the sample size is not determined correctly, the results will be inconclusive.

---

## [Author Response · Author response to Decision Letter 1]

26 Jun 2024

June 26, 2024 

Guilherme Antonio Moreira de Barros

Academic Editor

PLOS ONE

Re: [PONE-D-24-07292R1

CHRM3 (rs2165870) gene polymorphism is related to postoperative vomiting in female patients undergoing laparoscopic surgery.. Prospective observational study]

Dear Editor Guilherme Antonio Moreira de Barros, 

Thank you for your letter dated June 19, 2024, regarding our revised manuscript. We appreciate the helpful comments from you and reviewers. We have responded to these comments in detail below (the original criticisms from the reviewers are in italics). We revised the manuscript with tracked changes indicating all modifications which can be seen in the resubmitted version (please refer to the Word document).

Response to Editor Comments

The authors aimed to explore the relationship between the CHRM3 rs2165870 polymorphism and postoperative vomiting incidence in female patients who underwent laparoscopic surgery, resulting in the outcomes that the AA or A allele of the CHRM3 rs2165870 polymorphism elevated the risk of postoperative vomiting (AA versus GG; OR, 6.94; 95% CI, 1.49-32.46; P = 0.014; A versus G; OR, 2.52; 95% CI, 1.22-5.19; P = 0.012). The manuscript is largely well written and informative overall. However, there seem to be only a few minor concerns in this manuscript. The paper will be improved when the authors revise them according to the following comments.

Response: Thank you very much for taking the time to review our manuscript. We sincerely appreciate the hard work on this manuscript and professional academic feedback. We tried our best to improve the quality of our manuscript. 

Response to Reviewers’ Comments

1.All manuscript and supporting information: The gene names should be italic throughout the manuscript and supporting information.

Response: Thank you for this comment. We have changed all gene names to italic format. Please see the revised manuscript for details.

2. Results: “the three SNP loci of CHRM3” should be changed to “the three genotypes of the CHRM3 SNP”. [Page 11, Lines 204-205]

Response: Thank you for the reminder. We have made this change in the revised manuscript. 

3. Results: It would be better to describe the details about the results of association analysis between the postoperative analgesia-related indicators or adverse reaction rates between the KCNB2 rs349358 genotypes as well as CHRM3 rs2165870 genotypes. Even if such details are not presented in a main table, it would be better to be presented in a table in supporting information.

Response: Thank you for your comment. We have added relevant information about KCNB2 rs349358 genotypes and provided them as Supplementary Table1-3. 

4. Table 3: “P” in the “P < 0.05” should be italic.

Response: Thanks for the reminds. We have made this change in the revised manuscript. 

5. Abstract- Before the objectives, I missed a short background.

Response: We appreciate the feedback. In the abstract, we have added a brief background before the objectives. 

6. Explain how and why the sample size applied was stated. Explain how the study size was arrived at.

Response: We appreciate your remarks. The updated manuscript includes further information about the basis for the sample size estimate. Based on data from the literature (PMID: 18633022, PMID: 29843509), the baseline incidence rate of PONV after such surgeries is estimated to be 30-50%. Based on the mutation proportion of CHRM3 (PMID: 18633022), it is expected that the proportions of GG, GA, and AA genotypes are 45%, 45%, and 10%, respectively. Software G Power 3.1.9.2 was used to determine the sample size before this study with the following conditions: α = 0.05, β = 0.2, power value = 0.8, and the estimated odds ratio (OR) = 2.79. The total sample size required is 167.

7.Address possible confounding factors. And how it would alter the results.

Response: Possible confounding factors in this study including population stratification, linkage disequilibrium, environmental and lifestyle factors, measurement error and sample size etc. may all affect the results. Especially, the actual mutation rates were 67% (GG), 28.5% (GA), and 4.5% (AA) which is extremely different as initial estimated. Therefore, we recognized that increased sample size would enhance the statistical power.

8.Any missing data? And how do you address it?

Response: Furthermore, we missed detailed data related to KCNB2 in the initial submitted manuscript. It was not shown in the first version of the manuscript due to the lack of significant differences. And it has been included in the revised manuscript version(Table 2，Table 6 and Table 7).

9. Discuss the generalisability of the results.

Response: Thank you for this comment. We have added the generalizability of the results in the discussion. The results indicates that patients with specific gene polymorphism should be paid more attention and may be given certain medications to prevent the occurrence of PONV. 

10. The literature review needs to thoroughly cover other adverse events. The background does not provide enough justification as to why vomiting is considered more important than other adverse events.

Response: We appreciate your careful and considerate review remarks. In fact, all related adverse events beside nausea and vomiting were investigated and analyzed in the study and there is no significant difference among different genotypes of focused two genes. Vomiting was paid more attention in this study because it is the most often reported postoperative complication following laparoscopic surgery by patients. 

11. The sample size calculation lacks clarity. Stating the software used is not sufficient. It is necessary to clarify the assumptions and tests employed. Regarding the current results, it is difficult to determine whether the nonsignificant findings are due to insufficient statistical power.

Response: Thank you for the comment. We acknowledge that we should have included more information about the sample size calculation in the text. In the revised manuscript, we have added the relevant information on sample size calculation as blows. 

Based on data from the literature (PMID: 18633022, PMID: 29843509), the baseline incidence rate of PONV after such surgeries is estimated to be 30-50%. Based on the mutation proportion of CHRM3 (PMID: 18633022), it is expected that the proportions of GG, GA, and AA genotypes are 45%, 45%, and 10%, respectively. Software G Power 3.1.9.2 was used to determine the sample size before this study with the following conditions: α = 0.05, β = 0.2, power value = 0.8, and the estimated odds ratio (OR) = 2.79. The total sample size required is 167.

12. The statistical analysis used to evaluate the association between genotypes and adverse events appears to be cherry-picking, as the only significance is for vomiting. Nausea was not discussed in the background. P-values need to be adjusted for the 9 tests conducted (Table 3), and a marginally significant p-value of 0.022 will no longer be significant after adjustment.

Response: Thank you for the comments. Based on the data obtained in this study, the significant difference of adverse reactions only found in vomiiting and not in nausea among different CHRM3 genotypes. And there is not significant difference both in nausea and vomiting among KCNB2 rs349358 genotypes by statistical analysis. For CHRM3 genotypes, the final column of Table 3's P-value indicates the total P-value for the comparison of the three groups. The P-values for pairwise comparisons between the three groups for the individual indicator of vomiting are as follows.

Group P value

GG Vs.GA 0.204

GG Vs. AA 0.005

AA Vs.GA 0.102

Actually and same as your comments, the statistical difference was found between GG and AA genotype of gene CHRM3. The overall comparison p-value of vomiting incidence among the three groups is 0.022.

Moreover, the allelic model (A versus G; OR, 2.52; 95% CI, 1.22-5.19; P = 0.012) revealed that the A allele of the CHRM3 rs2165870 polymorphism raised the risk of postoperative vomiting in additional analysis. Therefore, despite the limited sample size, the current results show that the CHRM3 rs2165870 polymorphism's A allele increased the risk of postoperative vomiting, and this finding merits further investigation in clinical practice.

13. Further, is the marginal significance driven by the rare allele (AA, n=9)? The incident rates of vomiting for GG, GA and AA are 7%, 12% and 30% respectively and the significance is between GG and AA but not between GA and AA. Is the significance attributed to random chance? If the sample size is not determined correctly, the results will be inconclusive.

Response: We appreciate the insightful comments on the study's sample size. We are grateful for your concern and would like to go over this in more detail. As can be seen in the revised manuscript, the sample size was decided upon a thorough statistical power analysis which is sufficient to detect the expected effect size with a power of 0.8 at a significance level of 0.05. 

Initially, we roughly estimated that the GG, GA, and AA genotypes mutation rates is 45%, 45%, and 10%, respectively. The actual mutation rates, however, were found to be 4.5% (AA), 28.5% (GA), and 67% (GG). Exactly there are fewer mutations in the AA genotype than estimated. The incidence rates of vomiting for GG, GA, and AA are indeed 7%, 12%, and 30%, respectively. The observed significance between GG and AA, but not between GA and AA, suggests that the difference was not be solely due to random chance.

However, we acknowledged the possibility that a larger sample size maybe yield a greater statistical power or even different statistical results.

Thanks again, we appreciate your suggestion, and we have addressed the sample size limitations in the revised manuscript.

Sincerely,

Dr. Yu Jin on behalf of all authors.

Department of Anesthesiology, 

Chongqing Health Center for Women and Children, Women and Children’s Hospital of Chongqing Medical University, Chongqing, China, 401147#

---

## [Editor Report · Decision Letter 2]

7 Aug 2024

CHRM3 (rs2165870) gene polymorphism is related to postoperative vomiting in female patients undergoing laparoscopic surgery. Prospective observational study

PONE-D-24-07292R2

Dear Dr. Yu,

We’re pleased to inform you that your manuscript has been judged scientifically suitable for publication and will be formally accepted for publication once it meets all outstanding technical requirements.

Kind regards,

Guilherme Antonio Moreira de Barros, M.D., M.Sc., Ph.D

Academic Editor

PLOS ONE
---

## [Editor Report · Acceptance letter]

9 Aug 2024

PONE-D-24-07292R2 

PLOS ONE

Dear Dr. Yu, 

I'm pleased to inform you that your manuscript has been deemed suitable for publication in PLOS ONE. Congratulations! Your manuscript is now being handed over to our production team.

Kind regards, 

on behalf of

Dr. Guilherme Antonio Moreira de Barros 

Academic Editor

PLOS ONE